# PEBP4 Directs the Malignant Behavior of Hepatocellular Carcinoma Cells via Regulating mTORC1 and mTORC2

**DOI:** 10.3390/ijms23158798

**Published:** 2022-08-08

**Authors:** Qiongfeng Chen, Jingguang Jin, Wenhui Guo, Zhimin Tang, Yunfei Luo, Ying Ying, Hui Lin, Zhijun Luo

**Affiliations:** 1Jiangxi Province Key Laboratory of Tumor Pathogenesis and Molecular Pathology, Nanchang University, Nanchang 330031, China; 2Department of Pathology, School of Basic Sciences, Nanchang University, Nanchang 330031, China; 3Queen Mary School, Nanchang University, Nanchang 330031, China; 4Department of Pathophysiology, School of Basic Medical Sciences, Nanchang University, Nanchang 330031, China

**Keywords:** HCC, PEBP4, Akt, mTORC1, mTORC2

## Abstract

Phosphatidylethanolamine binding protein 4 (PEBP4) is an understudied multifunctional small protein. Previous studies have shown that the expression of PEBP4 is increased in many cancer specimens, which correlates to cancer progression. The present study explored the mechanism by which PEBP4 regulates the growth and progression of hepatocellular carcinoma cells. Thus, we showed that knockdown of PEBP4 in MHCC97H cells, where its expression was relatively high, diminished activities of serine/threonine protein kinase B (PKB, also known as Akt), mammalian target of rapamycin complex 1(mTORC1), and mTORC2, events that were not restored by insulin-like growth factor 1 (IGF-1). Conversely, overexpression of PEBP4 in MHCC97L cells with the low endogenous level yielded opposite effects. Furthermore, physical association of PEBP4 with Akt, mTORC1, and mTORC2 was observed. Interestingly, introduction of AktS473D mutant, bypassing phosphorylation by mTORC2, rescued mTORC1 activity, but without effects on mTORC2 signaling. In contrast, the effect of PEBP4 overexpression on the activity of mTORC1 but not that of mTORC2 was suppressed by MK2206, a specific inhibitor of Akt. In conjunction, PEBP4 knockdown-engendered reduction of cell proliferation, migration and invasion was partially rescued by Akt S473D while increases in these parameters induced by overexpression of PEBP4 were completely abolished by MK2206, although the expression of epithelial mesenchymal transition (EMT) markers appeared to be fully regulated by the active mutant of Akt. Finally, knockdown of PEBP4 diminished the growth of tumor and metastasis, whereas they were enhanced by overexpression of PEBP4. Altogether, our study suggests that increased expression of PEBP4 exacerbates malignant behaviors of hepatocellular cancer cells through cooperative participation of mTORC1 and mTORC2.

## 1. Introduction

Phosphatidylethanolamine binding protein 4 (PEBP4) is a multifunctional protein that belongs to a small family (PEBP1-4) in mammalian cells. PEBP1, also named RKIP (Raf kinase inhibitory protein), is well characterized by disrupting rapidly accelerated fibrosarcoma (Raf) and mitogen-activated protein (MEK) interaction via exclusive and competitive binding to either of them, leading to inhibition of signal transmission from Raf to MEK. However, the function of PEBP4 is poorly understood and involved in a broad range of biological processes [1,2,3,4,5,6]. PEBP4 is evolutionally divergent from the other three forms that share almost 80% identity and more than 90% similarity, suggesting that PEBP4 is functionally different from other forms. In fact, PEBP4 contains a signal peptide and has been detected in cell culture medium and body fluid such as serum, cerebrospinal fluid, and sperm plasma [1,3,4,7,8,9]. However, the biological relevance of its secretion is not defined. Many studies have focused on the function of PEBP4 inside cells. They have shown that PEBP4 enhances Akt activity while inhibiting extracellular regulated protein kinases (ERK) activation, possibly through a similar way to PEBP1 [10,11,12,13,14,15,16]. However, the argument on its inhibitory role in the activation of the ERK pathway sounds contradictory to the observations that the expression of PEBP4 is increased observed in many cancers.

Previous studies have indicated that PEBP4 promotes the progression of cancer [13,14,17,18]. Thus, it was reported that increased expression of PEBP4 was correlated with stages of lung squamous cell carcinoma; it was low at early (I and II) stages and increased at advanced (III and IV) stages and even greater in less differentiated cells [19]. Similar findings were reported in colorectal cancer and ovarian cancer [17], gliomas [20], meningioma [21], and breast cancer [22]. At cellular levels, it has been shown that knockdown of PEBP4 in cancer cells suppresses cell proliferation, migration, and invasion, leading to their apoptosis, whereas overexpression of PEBP4 induces opposite changes and confers resistance to chemotherapy and radiotherapy [12,13,14,16,17,23,24,25,26,27].

Mammalian Target of Rapamycin (mTOR) senses environmental cues and integrates inputs from growth factor, nutrient, energy, and stress to modulate anabolic processes [28]. mTOR plays a pivotal role in sustaining the growth and survival of cancer cells. Although mutation of mTOR is rarely reported in cancer, it is a converging point from many oncogenic nodes, such as mutations leading to the activation of the phosphatidyliositoal-3 kinase (PI3K)-Akt pathway and the rat sarcoma (Ras)-driven mitogen-activated protein kinase (MAPK) pathway. As a result, the activation of mTOR has been documented in up to 80% of human cancers [29]. In light of cellular functions and activation profiles, mTOR belongs to two complexes: mTORC1 and mTORC2. mTORC1 contains three core components: mTOR, mLST8 (mammalian lethal with SEC13 protein 8, also known as GβL), and Raptor (protein regulatory-associated protein of mTOR, a unique defining protein), while the core of mTORC2 is formed by mTOR, mLST8, and Rictor (rapamycin-insensitive companion of mTOR). mTORC1 is situated in the outer membrane of lysosomes and could be activated by growth factors and amino acids via different mechanisms. With regard to the activation of mTORC1 by growth factors, activated Akt phosphorylates tuberous sclerosis complexes 2 (TSC2) at multiple sites to dissociate TSC from the lysosomal surface and thus relieve inhibition of Ras homologue enriched in brain (Rheb) and mTORC1 [28]. The activated mTOR phosphorylates different downstream targets, such as ribosomal protein S6 kinase 1 (S6K1, also named p70S6K), 4E binding protein-1 (4EPBP1), hypoxia-inducible factor-1α (HIF-1α), and autophagy-initiating kinase 1 (ULK1), leading to promotion of protein synthesis, lipogenesis and angiogenesis, and inhibition of autophagy. The mechanism underlying mTORC2 activation is incompletely understood. It occurs at plasma membrane, where increased phosphatidylinositol (3,4,5)-trisphosphate (PIP3) binds to pleckstrin homology domain of mSin1 (stress-activated map kinase interacting protein 1). The pleckstrin homology domain of mSin1 autoinhibits mTORC2, which is released by binding to PIP3. Upon activation, mTORC2 phosphorylates hydrophobic motifs on Akt, protein kinase C (PKC), and serum glucose kinase (SGK1), members of AGC kinase family in a concert with phosphoinosi-tide-dependent kinase 1 (PDK1), which phosphorylates threonine in their activation loops [30]. Ultimately, mTORC2 plays an essential role in cell survival, proliferation, and cytoskeletal rearrangements.

Previously, we have shown that silencing PEBP4 suppresses phosphorylation of Akt at S473 but fails to inhibit ERK activation [1]. In the present study, we further characterized the effects of PEBP4 on the regulation mTORC1 and mTORC2. Our data revealed that knockdown of PEBP4 abolished activity of both mTORC1 and mTORC2 and that ectopic expression of Akt S473D mutant restored mTORC1 activity without an effect on mTORC2. We further showed that PEBP4 associated with components of both mTORC1 and mTORC2. Moreover, using hepatocellular carcinoma cell lines, we found that overexpression of PEBP4 enhanced malignant behaviors, including cell proliferation, migration, invasion, and metastasis, while its knockdown induced opposite changes. Collectively, our results suggest that PEBP4 overexpression in cancer cells promote their growth and progression through action on mTORC1 and mTORC2.

## 2. Results

### 2.1. The Effects of PEBP4 Expression on the Signal Transduction Pathways Downstream of mTORCs

We examined the expression profile of PEBP4 in various hepatic cell lines. As shown in Figure 1A,B, the expression of PEBP4 was low in immortalized normal hepatic cells (LO2) and hepatic-derived cancer cells of low malignancy (MHCC97L), while it was relatively high in more aggressive HCC (HCCLM3, MHCC97H) (*p* < 0.05 or *p* < 0.01). Therefore, we selected MHCC97H cells and then silenced PEBP4 with different shRNAs. The western blot data showed that sh-PEBP4-1, sh-PEBP4-2, and sh-PEBP4-4 had the best knockdown efficiency (Figure 1C,D). In the following experiments, we chose sh-PEBP4-1 to examine the knockdown effect on mTORC1 and mTORC2 activities. As parameters of mTORC1 activation, pS2448-mTOR and pS235/S236-S6 were used, while pS2481-mTOR, pT638/641-PKCα/β, pS422-SGK1, and pS473-Akt served as indicators for mTORC2 activation. In addition, pS227-RSK2 (Ribosomal S6 kinase 2) and pT308-Akt phosphorylated by PDK1 were also checked. Our results showed that phosphorylation of these sites was suppressed by PEBP4 knockdown, an event that was not restored by the treatment of IGF-1 (Figure 1E–J).

Next, we infected adenovirus expressing Akt S473D, a constitutively active mutant, and GFP as a control. The results revealed that the mutant derepressed the phosphorylation of mTORC1 substrates, as shown by increased phosphorylation of S2448-mTOR and S235/S236-S6, as well as T308-Akt and S227-RSK2, whereas pS2481-mTOR, pT638/641-PKCα/β, and pS422-SGK1 were still suppressed (Figure 2A–C). Interestingly, when pCMV6-PEBP4 was transfected into MHCC97L cells, MK-2206, an Akt specific inhibitor, diminished phosphorylation of S473 and T308 on Akt, S227-RSK2, S2448-mTOR, and S235/S236-S6, whereas pS2481-mTOR, pT638/641-PKCα/β, and pS422-SGK1 were unaltered (Figure 2D–I). All together, these results clearly indicated that PEBP4 played an important role in regulation of both mTORC1 and mTORC2 as well as PDK1.

### 2.2. PEBP4 Associates with Akt and mTORCs

We then assessed if PEBP4 associated with Akt, mTORC1, and mTORC2. First, we immunoprecipated Akt or mTOR from MHCC97H cells and blotted with antibodies against mTOR, Akt, and PEBP4. Our data showed that these three proteins were in the same complex (Figure 3A,B). Second, we performed immunoprecipitations of Akt and mTOR from MHCC97H/sh-PEBP4 cells and MHCC97H/sh-NC cells, respectively. The results revealed that knockdown of PEBP4 impaired the interaction between Akt and mTOR and even the association of Raptor and Rictor with mTOR (Figure 3C,D). Finally, as no good antibody to immunoprecipitated endogenous PEBP4 was available, we then expressed myc-PEBP4 in HEK293T cells and purified recombinant PEBP4 with myc antibody, followed by immunoblotting Akt, mTOR, Raptor and Rictor as well as myc epitope. Again, we found that PEBP4 associated with Akt, mTORC1, and mTORC2 (Figure 3E). Collectively, these results suggest that PEBP4 acts as a scaffold protein for Akt/mTOR signaling.

### 2.3. The Role of PEBP4 in Cell Proliferation and Tumor Growth

We first assessed the effect of PEBP4 shRNA on cell proliferation, colony formation, and tumor growth. As shown in Figure 4B–D, cell proliferation and colony formation were greatly suppressed when PEBP4 was silenced in MHCC79H cells as compared to control shRNA. We then employed Xenograft tumor model by injection of MHCC79H/sh-PEBP4 and MHCC79H/sh-NC cells, respectively. The results revealed that the growth of tumor significantly slowed down and at the endpoint, tumors from PEBP4 knockdown looked much smaller, and weight was greatly reduced.

We then examined the effect of PEBP4 overexpression on cell proliferation, colony formation, and tumor growth. To do this, we injected MHCC97L/PEBP4 cells and control cells with empty vector. As opposed to silencing PEBP4, overexpression of PEBP4 generated opposite effects, as shown by increased cell proliferation and colony formation (Figure 5B–D). When the cells were injected into right fossa axillaris, tumors grew very fast. At day 28, most of the tumors containing PEBP4 already reached maximal size allowed by animal ethics, whereas tumors without exogenous PEBP4 were still small (Figure 5E–G).

### 2.4. The Role of Akt in Mediating the Effect of PEBP4 on Proliferation, Migration, Invasion, and EMT of HCC

In light of our findings that PEBP4 regulated Akt activity, we asked if Akt mediated the effects of PEBP4 on proliferation, migration, invasion, and EMT. As shown in Figure 6, compared with adenovirus-directed expression of GFP, the PEBP4 knockdown-induced reduction of cell proliferation, migration, and invasion was significantly rescued by infection of adenovirus encoding Akt S473D, although not completely. Interestingly, the expression of EMT markers was almost completely recovered by Akt S473D, while it was inert to GFP. Next, we treated MHCC97L/PEBP4 with the Akt inhibitor MK2206. All of the parameters tested in Figure 6A–F were completely abrogated by the inhibitor (Figure 6G–L). Together, our results suggest that PEBP4 enhances the ability of cell proliferation, migration, and invasion, where Akt is necessary but not sufficient, although the effect of PEBP4 on expression of EMT markers is completely dependent on Akt. Of note, integrin was included in the WB analysis. It is involved in cancer metastasis, despite not being a direct marker of EMT.

### 2.5. PEBP4 Stimulates Metastasis of HCC

To assess the effect of PEBP4 on metastasis, we injected the cells with MHCC9H/sh-PEBP4 or MHCC9L/pCMV6-PEBP4 into tail veins of nude mice, as well as control cells with scrambled RNA or empty plasmids, respectively. As shown in Figure 7, nodules formed in the lung of sh-PEBP4 group were significantly fewer than those in the sh-NC group (*p* < 0.01), while the nodules formed in the PCMV6-PEBP4 group were more than in the PCMV6-NC group (*p* < 0.01). Histological examination showed obvious tumor invasion in the lung tissue.

## 3. Discussion

Previous studies have reported that PEBP4 is overexpressed in a variety of cancer specimens and the expression is correlated to cancer progression [13,14,17,18]. When PEBP4 is knocked down in cancer cells, the cell proliferation, migration, and invasion are reduced, and apoptosis is increased. The underlying mechanism is not clearly understood. In the present study, we chose hepatocellular cancer cells whose PEBP4 expression was not previously examined. Our results showed that silencing PEBP4 in relatively aggressive MHCC97H cells diminished activities of mTORC1 and mTORC2, while overexpression of PEBP4 in less aggressive MHCC97L cells yielded the opposite changes. Interestingly, introduction of Akt S473D mutant rescued mTORC1 activity without effects on mTORC2 when PEBP4 was knocked down. In contrast, the increased activity of mTORC1 but not mTORC2 was suppressed by MK2206, a specific inhibitor, in the cells overexpressing PEBP4. Moreover, physical association of PEBP4 with Akt, mTORC1, and mTORC2 was observed by immunoprecipitation. In addition, PEBP4 knockdown-engendered reduction of cell proliferation, migration, and invasion was partially rescued by Akt S473D but increases in these indices induced by overexpression of PEBP4 were completely abolished by MK2206, although the expression of EMT markers appeared to be fully dependent on Akt. Concomitantly, knockdown of PEBP4 suppressed the growth of tumor and metastasis derived from MHCC97H cells, whereas overexpression of PEBP4 enhanced the development of tumor and metastasis from MHCC97L cells. Altogether, our data indicate that PEBP4 participates in regulation of Akt/mTOR, leading to increased proliferation, migration, invasion, and metastasis of cancer cells.

mTOR exists in two complexes, mTORC1 and mTORC2, which are activated at different locations. While the activation of mTORC1 occurs in lysosomes in response to growth factors and nutrients, mTORC2 is activated at different membrane compartments, with the majority at plasma membrane [31,32]. In response to growth factors, Akt S473 is phosphorylated by mTORC2 at the plasma membrane, followed with phosphorylation of T308 by PDK1 through a PI3K-dependent mechanism [28]. The activated Akt then inactivates TSC2, leading to guanosine triphosphate (GTP) charging to Rheb and subsequent activation of mTORC1. As to how mTORC2 is targeted to plasma membrane, it is controversial. Liu et al. have reported that the activation of mTOR occurs after binding of PIP3, a PI3K product, to pleckstrin homology (PH) domain of Sin-1 which recruits mTORC2 to proximity to Akt [31]. In their model, the pleckstrin homology domain interacts with the mTOR catalytic domain, causing an inhibition of the kinase activity. In support of this, the authors showed that a mutation (D412G) in the PH domain of Sin-1 isolated from patients disabled its interaction with the catalytic domain of mTOR so as to activate mTORC2 and increase cell proliferation and tumor formation [31]. However, another study by Ebner et al. has shown that the PH domain of Sin-1 is necessary for membrane targeting, but which is PI3K-independent [32], as S473 phosphorylation is insensitive to PI3K inhibition when Akt is constitutively targeted to plasma membrane. Their studies suggest that a mechanism other than PIP3 is recruited to target mTORC2 to the plasma membrane.

Considering different locations of mTORC1 and mTORC2, an obvious question is how PEBP4 regulates their activities. Our study suggests that PEBP4 regulates the core enzyme mTOR by binding to mTOR complexes. This finding is reminiscent of a previous study by Li et al. [22], showing that hPEBP4 functioned as a scaffolding molecule and enhanced the association of Akt with selective catalytic reduction (Src) for full activation of Akt. It is possible that PEBP4 functions a scaffold to tether Akt or mTOR. However, at present we could not exclude another possibility that PEBP4 associates with Raptor and Rictor, which in turn influences its interaction with mTORC1 and mTORC2. To ascertain this, we will test the association between PEBP4 and mTOR when these those two proteins are knocked down.

It has been demonstrated that activated Akt positively regulates mTORC2 through forward feedback loop by phosphorylating Sin-1 at T86 [33,34]. This implies that active Akt enhances mTORC2 activity, which in turn potentiates phosphorylation of its downstream substrates, such as PKC and SGK1. In the present study, introduction of Akt S473D into MHCC97H cells containing PEBP4 knockdown did not restore the phosphorylation of PKC and SGK1. One explanation for this discrepancy might be that the positive feedback regulation only occurs when both mTORC2 and Akt are located at the plasma membrane, where phosphorylation of Akt at S473 and T308 takes place and Sin-1 might be phosphorylated by activated Akt as well. As the Akt S473D mutant is likely present in the cytoplasm where the constitutively active PDK1 is present in the resting state, Akt is functionally sequestered from membrane-associated Sin-1.

To our surprise, our results showed that PEBP4 augmented PDK1-induced phosphorylation of RSK2 S227, which appeared to be mediated by Akt. We do not believe this was an artifact for the following reasons: (1) knockdown of PEBP4 impaired RSK2 S227 phosphorylation, which was restored by Akt S473D, (2) increased RSK2 S227 phosphorylation upon overexpression of PEBP4 was suppressed by MK2206, and (3) in these two scenarios, mTORC2 activity toward PKC and SGK1 was not affected. Our findings are consistent with a previous report that MK2206 abrogated phosphorylation of Akt 473, and to a lesser extent, T308, although the study did not go into as much detail as ours [35]. It is not clear how PDK1 is regulated by active Akt. All kinases including Akt, and the downstream effectors mTORC1 and S6K1, could account for the regulation of PDK1. To our knowledge, it is rarely reported that PDK1 is positively regulated by kinase downstream of Akt, except a recent study by Jiang et al. [36] which shows that S6K1 phosphorylates PH domain of PDK1, followed by 14-3-3 binding and its dissociation from the plasma membrane, leading to the inhibition of Akt activation. Therefore, our findings add complexity to the regulation of PDK1, which warrants further investigation.

It is interesting to find that EMT was uncoupled with cell proliferation, migration, and invasion. Our results showed that downregulation of EMT markers by PEBP4 knockdown was completely reversed by Akt S473D, but the latter only partially restored cell proliferation, migration, and invasion. In contrast, overexpression of PEBP4 upregulated all these parameters, which were completely blunt by MK2206. Our explanation is that PEBP4 regulates both mTORC1 and mTORC2, and AktS473D only upregulates EMT makers but fails to activate mTORC2, which is required for cell proliferation and migration [28]. Therefore, we conclude that Akt-induced regulation of EMT is necessary but not sufficient to stimulate cell proliferation, migration, and invasion, which must also engage mTORC2.

As noted earlier, PEBP4 is a multifunctional protein that can function inside cells and may have a role outside because it can be secreted. Several studies have identified PEBP4 in body fluids such as serum and cerebrospinal fluid in cancer and brain injury, suggesting its role outside cells [1,3,4,7,8,9]. There are precedents for intracellular proteins being secreted and acting as paracrine factors, such as thioredoxin and high mobility group box 1 (HMGB1) [37,38]. It is tempting to speculate that low levels of PEBP4 mainly function inside the cells and part is secreted if it is overexpressed, for example, in advanced stages of cancer. It will be intriguing to investigate the functional role of PEBP4 as a secreted molecule in addition to elucidate the mechanisms by which PEBP4 regulates intracellular signaling mediated by PI3K, PDK1, Akt, and mTOR.

In the present study, we selected two hepatocellular carcinoma cell lines, MHCC97H with relatively high levels of PEBP4 and MHCC97L at its low levels, and then manipulated the expression levels of PEBP4. Our data clearly demonstrated that PEBP4 regulated mTORC1 and mTORC2. Furthermore, we were able to show the physical association among these proteins. The regulation of Akt and mTOR concurs with biological behaviors of PEBP4, such as stimulation of cell proliferation, migration, and metastasis of HCC cells. Therefore, our data from mechanistic points reinforces important roles of PEBP4 in cancer progression.

## 4. Materials and Methods

### 4.1. Materials and Reagents

IGF-1, BCA Kit, penicillin/streptomycin mix, and Puromycin were purchased from Beijing Soleibao Company (Beijing, China). Akt specific inhibitor (MK-2206) was from MedChem Express (Monmouth, NJ, USA). Adenovirus encoding the human Akt S473D mutant carboxyterminally tagged 3 copies of flag epitope (1.2 × 10^10^ pfu/mL) and the GFP adenovirus (1.5 × 10^10^ pfu/mL) were from WZ Biosciences Inc. (Jinan, Shandong, China). Lipofectamine 3000 was from Thermo Fisher Scientific (Waltham, MA, USA). G418 was from Gibco (Grand Island, New York, NY, USA). Transwell chamber and matrigel were from BD Biosciences (San Jose, CA, USA). Protein A/G-agarose was from Santa Cruz Biotechnology (Santa Cruz, CA, USA). CCK8 was from AbMole (Houston, TX, USA). Myc-agarose was Sigma-Aldrich (St. Louis, MI, USA). HyClone Dulbecco’s Modified Eagle’s medium (DMEM) and fetal bovine serum (FBS) were purchased from Cytiva (Marlborough, MA, USA). All antibodies with catalog number and manufactures are listed in Table 1.

### 4.2. Cell Culture

Immortalized hepatic cell line (LO2) and 5 hepatocellular carcinoma cell lines (MHCC97L, HepG2, SMMC-7721, MHCC97H, HCCLM3) were purchased from Guangzhou Jinio Biotechnology Co., Ltd. (Guangzhou, China). All cells were cultured in 10% FBS-DMEM supplemented with 1% penicillin/streptomycin in a tissue culture incubator filled with 5% CO_2_ and saturated humidity at 37 °C.

### 4.3. Virus Production and Transfection

The lentiviral vector bearing short hairpin RNA for human PEBP4, PEBP4 overexpression plasmid (pCMV6-PEBP4), and control vectors were purchased from Beijing OriGene Technology Co., Ltd. (Beijing, China). ShRNA sequences were listed in Table 2. Lentivirus was prepared in HEK293T cells, as described previously [39]. The hepatocellular carcinoma MHCC97H cells were infected with lentivirus expressing PEBP4 shRNA or scrambled shRNA and selected with puromycin (2 µg/mL). Stable expression was achieved by transfection of MHCC97L cells with pCMV6-PEBP4 or control plasmid using Lipofectamine 3000 according to the manufacturer’s protocol and selection with G418 (0.9 mg/mL). Transient expression of N-myc-PEBP4 was conducted in HEK 293T cells and the cell extracts were prepared after 48 h.

### 4.4. Adenovirus Infection

The cells (MHCC97H/sh-NC, MHCC97H/sh-PEBP4) were infected with adenovirus at different doses (2.4, 7.2, and 12 pfu/cell) for 36 h prior to experiment.

### 4.5. Western Blot and Immunoprecipitation

Western blot was performed according to the standard protocol. Briefly, cell extracts (20 µg) were prepared in RIPA buffer containing protease inhibitor cocktail and protein phosphatase inhibitors. Protein concentrations were assayed by the BCA method, and equal amounts of proteins were separated onto SDS-PAGE and transferred to PVDF membranes. Then, the membranes were blocked with 5% non-fat milk at room temperature and subsequently incubated with primary antibodies and HRP-conjugated second antibodies. Specific bands were visualized after staining with luminescent chemicals. Densitometric units were determined by Image J software.

Co-immunoprecipitation was performed according to Zang et al. [40]. In brief, 15 μL of protein A/G-agarose were preincubated with 0.5~1 µg antibodies and incubated with cell extracts (500 µg) prepared in a cell lysis buffer (20 mM Tris-Cl, pH 8.0, 137 mM NaCl, 2 mM EDTA, 1% NP-40, protease inhibitors, and phosphatase inhibitors). After 4 to 16 h, the samples were precipitated by short spin and washed with lysis buffer. The immunoprecipitates were analyzed by Western blot.

### 4.6. Cell Proliferation Assay

Cell proliferation assay was performed as described previously [41]. Briefly, cells (6000 cells/well) were seeded on a 96-well plate in DMEM supplemented with 10% FBS. Cell proliferation was examined using Cell Counting kit (CCK-8) on microplate reader (SpectraMax Paradigm, Silicon Valley, CA, USA) at 0, 24, 48 h, according to the manufacturer’s instructions.

### 4.7. Cell Migration and Invasion Assays

The assay was carried out as described previously [42]. For cell migration, cells (1 × 10^4^) were suspended in 100 μL serum free DMEM and seeded onto the upper well of a transwell chamber, and 10% FBS-DMEM was added to fill the lower well. After incubation at 37 °C for 36 h, the cells on the lower surface of the chamber were fixed and stained. Ten random fields were photographed under microscope. For cell invasion, 100 μL matrigel was added to the upper well of a transwell chamber and 10^4^ cells seeded in 100 μL serum-free DMEM medium, followed by filling 500 μL 10% FBS-DMEM to the lower well. The next steps were the same as transwell migration assay.

### 4.8. Colony Formation Assay

Colony formation assay was performed as described by Wang et al. [43]. Briefly, cells (500 cells/well) were evenly seeded in a 6-well plate. Ten days after incubation, the cell colonies were washed with cold PBS twice, fixed with 4% paraformaldehyde at room temperature for 60 min, and stained with 0.1% crystal violet diluted in PBS at room temperature for 20 min. The colonies containing more than 50 cells were included for counting.

### 4.9. Hematoxylin and Eosin (H&E) Staining

H&E staining was carried out according to Hu et al. [44]. In short, tissues were fixed in 10% formaldehyde for 24 and embedded in paraffin, followed by sectioning (5 µm), patching, hydration, and staining with hematoxylin and eosin (H&E). Finally, images of different magnifications were captured by microscopic photography.

### 4.10. Xenograft Tumor Model

Animal protocols were approved by the Animal Protection Committee of Nanchang University Jiangxi Medical College (The protocol number: NCDXSYDWFL-2015097).

Male BALB/c nude mice at 5 weeks old were purchased from Hangzhou Ziyuan Laboratory Animal Technology Co, Ltd. (Hangzhou, China), acclimatized in the animal center for one week, and randomly divided into groups as indicated in the Results (n = 5). The cells (MHCC97H/sh-NC, MHCC97H/sh-PEBP4, MHCC97L/pCMV6-NC, and MHCC97L/pCMV6- PEBP4, respectively) were injected (5 × 10^6^ cells/100 µL/spot) into right axillaris of mice [45]. When tumors were palpable, the volumes were measured every other day according to the formula of length × width^2^ × 0.5. Finally, the mice were sacrificed at day 20 (for MHCC97H/sh-NC and MHCC97H/sh-PEBP4) or day 28 (MHCC97L/pCMV6-NC, MHCC97L/pCMV6-PEBP4) and tumors were collected for measurement of tumor weight and morphological analysis.

### 4.11. Lung Metastasis Assay

The cells (MHCC97H/sh-NC, MHCC97H/sh-PEBP4, MHCC97L/pCMV6-NC, and MHCC97L/pCMV6-PEBP4, respectively) were suspended in 100 μL PBS (5 × 10^6^ cells) and injected into tail veins of male nude mice [45]. The mice were sacrificed after 10 weeks and metastatic nodules in the lungs examined.

### 4.12. Statistical Analysis

All data were expressed as mean±standard error of mean (SEM) and analyzed using the statistical package for the social sciences (SPSS) 17.0 software (SPSS, Chicago, IL, USA). For comparing the differences between groups at the same time, one-way analysis of variance or student-*t* test was used, while for comparing differences in multiple factors (e.g., multiple groups and points), two-way ANOVA was used. In all experiments, *p* < 0.05 indicates significant difference.

## Figures and Tables

**Figure 1 ijms-23-08798-f001:**
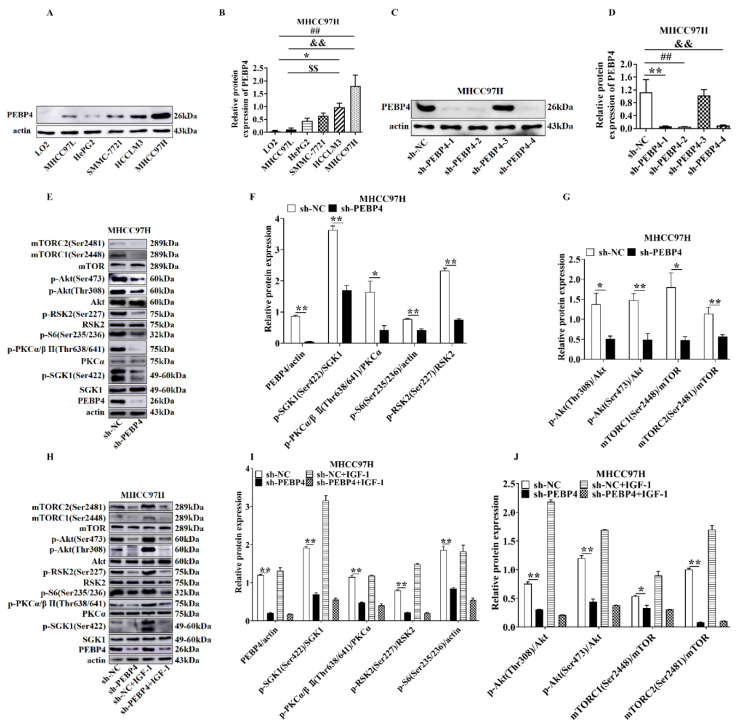
The effect of PEBP4 knockdown on the mTORCs-mediated signaling pathways. (**A**) Western blot (WB) was performed using different HCC and immortalized normal hepatocytes as indicated. (**B**) Scan densitometric ratio of PEBP4 to actin. (**C**,**D**). MHCC97H cells were infected with sh-NC and different PEBP4 shRNA1-4. Cell extracts were blotted (**C**) and quantified by scan densitometry (**D**). (**E**) Extracts of MHCC97H cells with sh-PEBP4 and sh-NC were blotted with antibodies. (**F**,**G**) Relative scan densitometric units were determined by the ratio of phospho-proteins to their cognate total proteins or actin for (**E**). (**H**) MHCC97H cells with sh-PEBP4 or sh-NC were treated with or without IGF-1 (100 µg/mL) for 30 min and WB was performed using antibodies. (**I**,**J**) Relative scan densitometric units were determined for H. All graphs represent averages of three independent experiments (mean ± SEM). * *p* < 0.05, ** *p* < 0.01, ^##^ *p* < 0.01, ^&&^ *p* < 0.01, ^$$^ *p* < 0.01.

**Figure 2 ijms-23-08798-f002:**
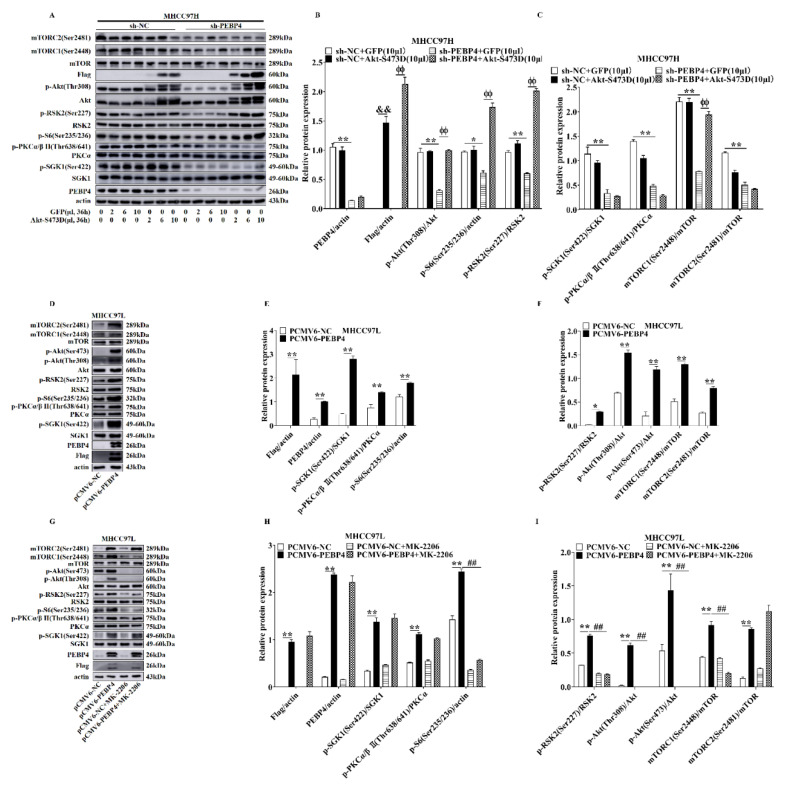
Akt in mediating the effect of PEBP4 on mTORC1, but not mTORC2. (**A**) The MHCC97H cells containing sh-PEBP4 or sh-NC were infected with adenovirus encoding the active Akt1 mutant (Akt-S473D) or GFP as a control at different doses and Western blots carried out using antibodies, as indicated. (**B**,**C**) The immunoblotted bands were analyzed by densitometry, as in Figure 1. (**D**,**G**) The MHCC97L cells containing pCMV6-PEBP4 or pCMV6-NC were treated with or without MK-2206 (5 µM, 24 h), an Akt specific inhibitor, and extracted blotted with antibodies. (**E**,**F**,**H**,**I**) Quantification of bands in (**D**,**G**) was performed. Results were presented as mean ± SEM (n = 3). Significant differences were examined by student’s *t*-test. * *p* < 0.05, ** *p* < 0.01, ^##^ *p* < 0.01, ^&&^ *p* < 0.01, ^ϕϕ^
*p* < 0.01.

**Figure 3 ijms-23-08798-f003:**
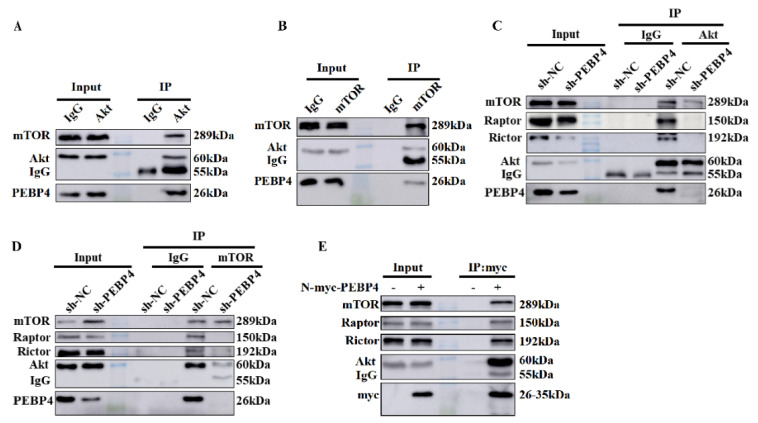
PEBP4 interacts Akt and mTOR. (**A**,**B**) Extracts of MHCC97H cells were immunoprecipated with anti-Akt antibodies (**A**) or anti-mTOR (**B**) antibodies and blotted with antibodies as indicated. (**C**,**D**) Extracts of MHCC97H/sh-PEBP4 and MHCC97H/sh-NC cells were immunoprecipitated with anti-Akt (**C**) or anti-mTOR antibodies (**D**), followed by WB. (**E**) HEK293T cells were transiently transfected with plasmid encoding PEBP4 aminoterminally tagged with myc epitope (N-myc-PEBP4) and the recombinant PEBP4 was immunoprecipitated by anti-myc antibody, followed by WB.

**Figure 4 ijms-23-08798-f004:**
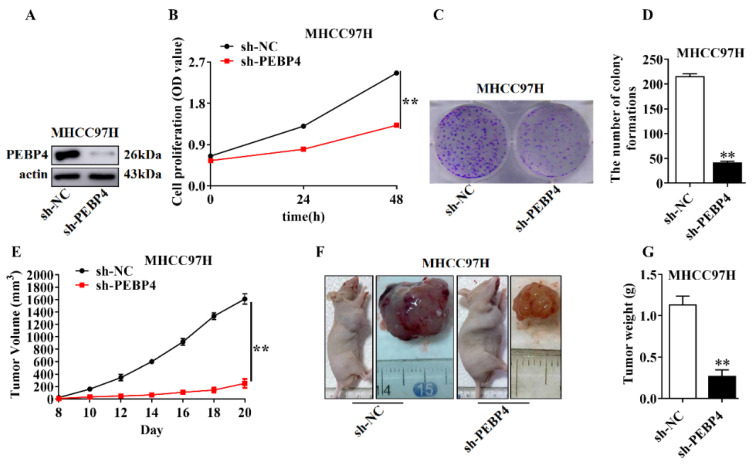
Knockdown of PEBP4 suppresses the growth of HCC. (**A**–**D**) MHCC97H cells containing sh-PEBP4 or sh-NC were examined by WB (**A**), cell proliferation (**B**), and colony formation (**C**,**D**). Colonies in triplicate were counted (**D**). (**E**) BALB/C nude mice (n = 5) were injected into right fossa axillaris with the cells as indicated in (**A**) and tumor volumes measured. (**F**,**G**) Mice were sacrificed after 20 days and tumors were weighed (**G**). Representative images of mice and isolated tumors (**F**) were presented. All quantitative measurements were presented as mean ± SEM and statistical significance was tested, ** *p* < 0.01.

**Figure 5 ijms-23-08798-f005:**
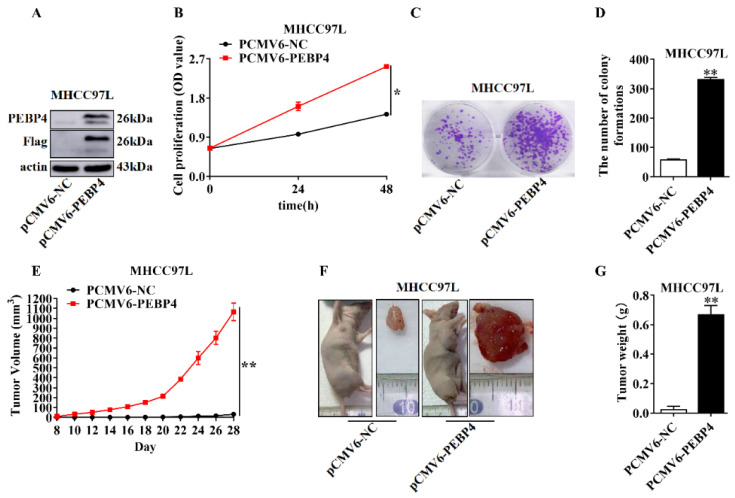
Overexpression of PEBP4 promotes the growth of HCC. (**A**) MHCC97L cells were transfected with pCMV6-PEBP4 or pCMV6-NC (empty vector) and then examined by WB. (**B**) Cell proliferation. (**C**) Colony formation. Colonies in triplicate were counted (**D**). (**E**) BALB/C nude mice (n = 5) were injected into right fossa axillaris with the cells indicated in (**A**), and tumor volumes measured. (**F**,**G**) Mice were sacrificed after 28 days, and tumors were weighed (**G**). Representative images of mice (right fossa axillaris) and isolated tumors (**F**) were presented. Statistical analysis was performed as for Figure 4. * *p* < 0.05, ** *p* < 0.01.

**Figure 6 ijms-23-08798-f006:**
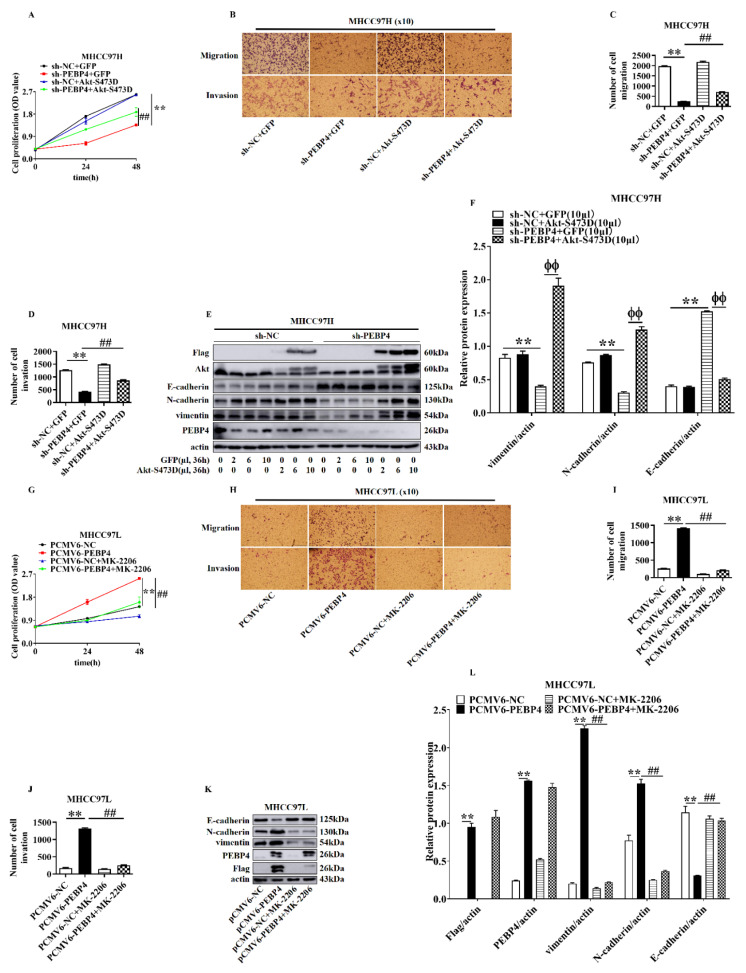
The role of Akt in mediating the effect of PEBP4 on proliferation, migration, invasion, and EMT markers of HCC. (**A**–**D**) MHCC97H cells containing sh-PEBP4 or sh-NC were infected with adenovirus expressing Akt S473D or GFP. Cell proliferation (**A**), migration and invasion were assayed (**B**) and quantified (**C**,**D**). (**E**,**F**) EMT markers were examined by Western blot € and WB bands semi-quantified as opposed to actin (**F**). (**G**–**L**) MHCC97L cells transfected with pCMV6-PEBP4 or the empty vector pCMV6-NC were treated with MK2206 or DMSO, cell proliferation (**G**), cell migration, and invasion were examined (**H**) and quantified (**I**,**J**). EMT markers were examined by WB (**K**) and bands were quantified against actin (**L**). All experiments were repeated three times. Data were presented as mean ± SEM (n = 3), ** *p* < 0.01, ^##^
*p* < 0.01, ^ϕϕ^
*p* < 0.01.

**Figure 7 ijms-23-08798-f007:**
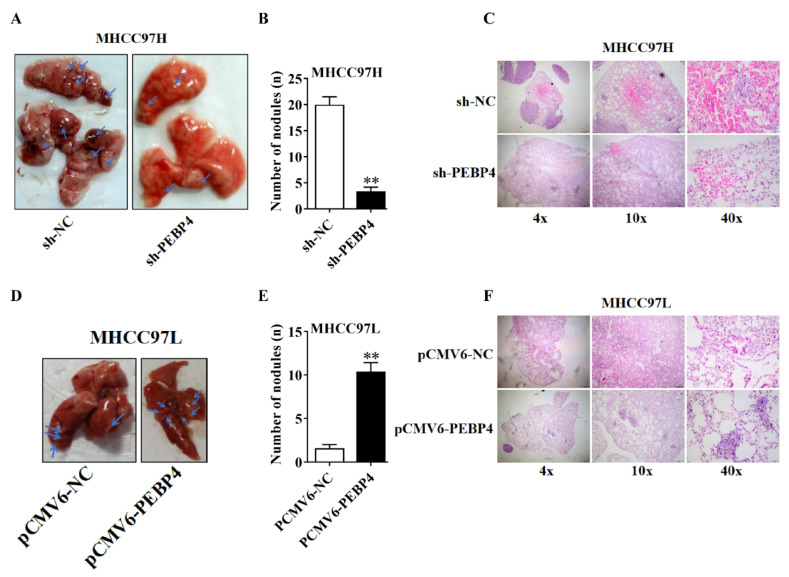
The role of PEBP4 in regulation of metastasis of HCC. (**A**–**C**) MHCC97H/sh-NC or MHCC97H/sh-PEBP4 cells were injected into tail veins of nude mice (n = 5/group) and 10 weeks later, lungs were dissected and photographed. The representative images of lung appearance were shown (**A**). The metastatic nodules in the lung were counted and plotted (**B**). Representative hematoxylin and eosin (H&E) stained images of the lung were shown (magnification: ×40, ×100, ×400) (**C**). (**D**–**F**). MHCC97L/pCMV6-PEBP4 and MHCC97L/pCMV6-NC cells were injected, and lung metastasis was assessed similarly to (**A**–**C**). Arrows point to metastatic nodules. Data were presented as mean ± SEM (n = 5), ** *p* < 0.01.

**Table 1 ijms-23-08798-t001:** List of antibodies used in this article.

Target	Catalog No.	Source
β-actin	TA-09	Zhongshan Golden Bridge (Beijing, China)
PEBP4	ab139074	Abcam (Cambridge, UK)
p-SGK1 (Ser422)	ab55281	Abcam
Akt	ab32505	Abcam
Rictor	ab70374	Abcam
SGK1	28454-1-AP	Proteintech (Wuhan, China)
PKCα	21991-1-Ig	Proteintech
RSK2	23762-1-AP	Proteintech
mTOR (2448)	67778-1-Ig	Proteintech
E-cadherin	00077111	Proteintech
N-cadherin	00059200	Proteintech
Vimentin	00066574	Proteintech
myc	60003-2-Ig	Proteintech
p-PKCα/β (Thr638/641)	9375	Cell Signaling Tech. (Danvers, MA, USA)
p-S6 (Ser235/236)	4858	Cell Signaling Tech.
p-RSK2 (Ser227)	3556	Cell Signaling Tech.
p-Akt (Thr308)	13038s	Cell Signaling Tech.
p-Akt (Ser473)	4060s	Cell Signaling Tech.
mTOR	2983	Cell Signaling Tech.
integrinβ-1	34971T	Cell Signaling Tech.
Raptor	2280	Cell Signaling Tech.
mTOR (2481)	PA5104898	Thermo scientific (Waltham, MA, USA)
Flag	T0053	Affinity (Miami, FL, USA)
Rabbit IgG	A7016	Beyotime (Shanghai, China)

**Table 2 ijms-23-08798-t002:** shRNA sequences.

Plasmid	Sequences
sh-PEBP4-1	ACCTCCTGGATGGAGCCGATAGTCAAGTT
sh-PEBP4-2	TGGCTTCCATCGCTACCAGTTCTTTGTCT
sh-PEBP4-3	GTTGGACAATGAGGCTGGTTACAGCAGCA
sh-PEBP4-4	GGACAGATTTCTGAACCGCTTCCACCTGG

## Data Availability

All data generated during this study are included in this published article, and the datasets described in this study can be acquired from the corresponding author upon request.

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
