# Peer review of "PEBP4 Directs the Malignant Behavior of Hepatocellular Carcinoma Cells via Regulating mTORC1 and mTORC2"

_ijms, 2022, doi:10.3390/ijms23158798_

Round 1
Reviewer 1 Report
The authors have expanded on their earlier study of PEBP4 where they analysed its secretion and regulation of Akt. In this new work they have explored the signalling consequences of high and low PEBP4 expression. They have found links with mTORC1, mTORC2 and Akt signalling. Manipulation of PEBP4 levels affects the migration, invasion and proliferation of hepatocellular carcinoma cells in vitro and in vivo. This is a thorough study of PEBP4 in the hepatocellular carcinoma setting and provides strong evidence for the involvement of PEBP4 in the Akt/mTOR signalling axis.
Major Points:
1. I feel the work has been suitably controlled and generally the results are appropriately discussed. However, Figure 1, 2 and 6 are currently too small to be easily studied and the size of each panel needs to be larger for a final version. It is difficult to make out the narrow bars in some graphs e.g. Figure 2B and C. Perhaps the authors could consider just showing one dose of GFP/Akt-S473D in the graphs, rather than all doses (but keep all doses in the blots so that readers can see the overall pattern) to help the readability of the figure.
2. In the discussion, the authors state “Our study suggests that PEBP4 regulates the core enzyme mTOR by direct binding, as our results showed that PEBP4 associated with mTOR and in the absence of PEBP4, association of mTOR with Raptor and Rictor was reduced.” (Lines 311-314). However, given the evidence presented I don’t feel they can claim direct binding as there are no Raptor or Rictor knockdown experiments to see if PEBP4 and mTOR still associate in the absence of these complex components which are known to be involved in binding other proteins to the complex. Equally they didn’t mutate any key binding domains of mTOR and show loss of PEBP4 binding. Therefore I think the wording in the conclusion should be modified to indicate that PEBP4 binds the complex, but further work is needed to see where exactly it binds.
Minor points:
1. Is ‘hepatocytes’ (line 118) the correct term to use when the cells are cancer derived? Would hepatic cell lines or hepatic-derived cell lines be more accurate?
2. The Ser2448 blot in Fig2A looks much more pixelated than the others as if the contrast, etc has been substantially modified. I think this blot should be replaced.
3. There are no error bars visible on Figure 4B – was this expt repeated? If so, were the replicates so tight as to make the error bars too small to see?
4. The arrows are not very clear in Figure 7A (and to some extent D). These should be made clearer.
5. The wording in lines 173-174 should be checked – no good antibody was available (not unavailable)
Reviewer 2 Report
The present study analyzed the functional role of Phosphatidylethanolamine binding protein 4 (PEBP4) in hepatocellular carcinoma (HCC) cell lines. The authors showed the expression of PEBP4 is related to the regulation of mTORC2 expression and the activation of its downstream signaling pathway. The protein interaction of PEBP4 with Akt or mTORCs was demonstrated. In addition, the expression of PEBP4 was related to the growth and metastasis of MHCC97 cells. The reviewer considers the present study is well-constructed and the amount of analysis is appropriate. The reviewer considers the present manuscript needs proofreading to revise some typographical errors.
Author Response
Comments and Suggestions for Authors:The present study analyzed the functional role of Phosphatidylethanolamine binding protein 4 (PEBP4) in hepatocellular carcinoma (HCC) cell lines. The authors showed the expression of PEBP4 is related to the regulation of mTORC2 expression and the activation of its downstream signaling pathway. The protein interaction of PEBP4 with Akt or mTORCs was demonstrated. In addition, the expression of PEBP4 was related to the growth and metastasis of MHCC97 cells. The reviewer considers the present study is well-constructed and the amount of analysis is appropriate. The reviewer considers the present manuscript needs proofreading to revise some typographical errors.
Response: We are grateful to this reviewer for the positive comments. We had carefully proofread the manuscript to minimize typographical, grammatical, and bibliographical errors. All changes made in the revision are tracked such that they can be easily identified.